# Taxonomic Evaluation of the *Heyndrickxia* (Basonym *Bacillus*) *sporothermodurans* Group (*H.*
*sporothermodurans*, *H.*
*vini*, *H. oleronia*) Based on Whole Genome Sequences

**DOI:** 10.3390/microorganisms9020246

**Published:** 2021-01-26

**Authors:** Gregor Fiedler, Anna-Delia Herbstmann, Etienne Doll, Mareike Wenning, Erik Brinks, Jan Kabisch, Franziska Breitenwieser, Martin Lappann, Christina Böhnlein, Charles M. A. P. Franz

**Affiliations:** 1Department of Microbiology and Biotechnology, Max Rubner-Institut (MRI), Federal Research Institute of Nutrition and Food, Hermann-Weigmann-Straße 1, 24103 Kiel, Germany; Anna-Delia.Herbstmann@bfr.bund.de (A.-D.H.); erik.brinks@mri.bund.de (E.B.); jan.kabisch@mri.bund.de (J.K.); christina.boehnlein@mri.bund.de (C.B.); 2Chair of Microbial Ecology, ZIEL—Institute for Food & Health, Technical University of Munich, Weihenstephaner Berg 3, 85354 Freising, Germany; etienne.doll@tum.de (E.D.); Mareike.Wenning@lgl.bayern.de (M.W.); 3Bavarian Health and Food Safety Authority, Veterinärstraße 2, 85764 Oberschleißheim, Germany; 4Tetra Holdings GmbH, Untere Waldplätze 27, 70569 Stuttgart, Germany; Franziska.Breitenwieser@tetrapak.com (F.B.); Martin.Lappann@tetrapak.com (M.L.)

**Keywords:** *Bacillus*, sporothermodurans, *oleronia*, vini, phylogeny, spores, UHT-milk, diversity, core genome, Single Nucleotide Polymorphisms (SNPs), *Heyndrickxia*

## Abstract

The genetic heterogeneity of *Heyndrickxia sporothermodurans* (formerly *Bacillus*
*sporothermodurans)* was evaluated using whole genome sequencing. The genomes of 29 previously identified *Heyndrickxia*
*sporothermodurans* and two *Heyndrickxia vini* strains isolated from ultra-high-temperature (UHT)-treated milk were sequenced by short-read (Illumina) sequencing. After sequence analysis, the two *H. vini* strains could be reclassified as *H. sporothermodurans*. In addition, the genomes of the *H.*
*sporothermodurans* type strain (DSM 10599^T^) and the closest phylogenetic neighbors *Heyndrickxia*
*oleronia* (DSM 9356^T^) and *Heyndrickxia vini* (JCM 19841^T^) were also sequenced using both long (MinION) and short-read (Illumina) sequencing. By hybrid sequence assembly, the genome of the *H. sporothermodurans* type strain was enlarged by 15% relative to the short-read assembly. This noticeable increase was probably due to numerous mobile elements in the genome that are presumptively related to spore heat tolerance. Phylogenetic studies based on 16S rDNA gene sequence, core genome, single-nucleotide polymorphisms and ANI/dDDH, showed that *H. vini* is highly related to *H. sporothermodurans*. When examining the genome sequences of all *H.*
*sporothermodurans* strains from this study, together with 4 *H. sporothermodurans* genomes available in the GenBank database, the majority of the 36 strains examined occurred in a clonal lineage with less than 100 SNPs. These data substantiate previous reports on the existence and spread of a genetically highly homogenous and heat resistant spore clone, i.e., the HRS-clone.

## 1. Introduction

*Heyndrickxia* (*H.*) *sporothermodurans, H. oleronia* and *H. vini* are gram-positive, rod-shaped and endospore-forming bacteria [1,2,3]. These three species are closely related, with their nearest phylogenetic neighbors being *Margalitia shackletonii* and *Bacillus* (*B.*) *acidicola* [4]. Recently a new classification of *Bacillus* species was published by Gupta et al. (2020), which proposes a reclassification and renaming of, among others, *B. sporothermodurans* [5]. Based on this valid taxonomic reclassification, the new names listed in Table 1 were used in this study.

*Heyndrickxia sporothermodurans* is a non-pathogenic bacterium whose highly-heat-resistant spores (HRS) are able to survive the food sterilization process [7]. Their presence in milk products does not lead to visible spoilage, but it leads to breaching of commercial sterility. Since the first report of *Heyndrickxia* contamination of ultra-high-temperature (UHT)- and sterilized milk in Italy, Austria and Germany [8] and the description of the new species *H. sporothermodurans* (*Bacillus sporothermodurans* sp. nov.) [1], many countries reported the isolation of *H. sporothermodurans* from UHT milk, feed and food [9,10,11,12,13,14,15,16]. Various authors discussed the heterogeneity of *H. sporothermodurans* strains and the potential spread of a specific clonal linage, the so-called HRS-clone [17,18]. However, methods like ribotyping and fingerprinting methods show a higher genetic diversity than previously assumed [4,14].

*Heyndrickxia oleronia* was described in 1995 by Kuhnigk et al. [2]. It was first isolated from the termite *Reticulitermes santonensis* (Feytaud) [2], but later also from raw milk and feed [4,9,13,19]. *Heyndrickxia oleronia* is phylogenetically closely related to *H. sporothermodurans* and was found to be difficult to distinguish from the latter on the basis of biochemical and physiological characteristics solely [4]. The draft genome of *H. oleronia* was published 2017 by Owuso-Darko et al. [20] with more than 500 contigs after sequence assembly. This underlines the difficulty of assembling *Heyndrickxia* species genomes using a sequencing technique such as Illumina MiSeq sequencing that generates short-reads only. *Heyndrickxia vini* was isolated from an alcohol fermentation pit mud and was described in 2016 by Ma et al. [3]. This *Heyndrickxia* sp. showed the closest relatedness to *H. sporothermodurans* and *H. oleronia* type strains, with 98.4% and 97.2% sequence similarity of the 16S rRNA gene. Classical DNA-DNA hybridization values were 33.3% and 42.8% between *H. vini* vs. *H. sporothermodurans* and *H. vini* vs. *H. oleronia* [3], respectively, indicating that these are different species. Based on further differences in the morphological, physiological, and biochemical characteristics, when compared to both *H. sporothermodurans* and *H. oleronia,* the strain was then described as a new species, namely *H. vini* [3], despite the lack of whole genome sequencing data.

Whole genome sequencing offers valuable support in understanding the genetic background and phylogenetic diversity of bacteria. It currently provides the best possibility to clarify genomic relationships with the highest discriminatory power [21]. At the time of writing, the genome of the *H. vini* type strain was not published and the genome sequences of only five *H. sporothermodurans* strains were available, i.e., those of the type strain DSM 10599^T^, strain BR3 (also named BR12), strain SAD, strain SA01 and B4102 [15,22]. Their assemblies consist of 100–800 contigs, which is a relatively high number when compared to other bacterial species. Usually, depending on the species, contig numbers between 15 and 200 contigs are produced by short-read sequence assembling. For example, published contig numbers are 15 to 30 contigs for *Campylobacter jejuni* [23], 50 to 100 for *Salmonella enterica* [23], 10 to 30 for *Listeria monocytogenes* [23,24], 80 to 100 for *Escherichia coli* [24,25] and 50 to 100 contigs for genomes of the *B. cereus* group [26]. The aim of this study was to increase the availability of *H. sporothermodurans* genomic DNA sequences by sequencing an additional 29 strain genomes of *H. sporothermodurans*. These isolates represent problematic spoilage bacteria in the food sector and were isolated from UHT milk. In addition, the combination of short- and long-read sequencing should significantly improve the genome quality of the type strains of *H. sporothermodurans*, *H. oleronia* and *H. vini*. Based on the genomic sequences, this study furthermore aimed to evaluate the genetic diversity of the *H. sporothermodurans* group, consisting of the closely related species of *H. sporothermodurans*, *H. oleronia* and *H. vini,* and to determine phylogenetic relationships between *H. sporothermodurans* strains.

## 2. Materials and Methods

### 2.1. Strains and Media

Thirty *H. sporothermodurans* strains (29 isolates and the type strain DSM 10599^T^), three *H. vini* strains (two presumptively identified isolates and the type strain JCM 19841^T^) and the *H. oleronia* type strain DSM 9356^T^ (Table 2) were obtained from the German Collection of Microorganisms and Cell Cultures (DSMZ, Braunschweig, Germany) or were kindly provided by the ZIEL—Institute for Food & Health (TU Munich, Germany) and by Tetra Holdings GmbH (Aseptic and Filling Solutions, Stuttgart, Germany). The *H. vini* strains M5352-102 and M5572-6 were previously identified on the basis of 16S rRNA gene sequencing in the absence of whole genome sequencing data. All strains were cultivated aerobically in brain heart infusion (BHI) broth (Carl Roth, Karlsruhe, Germany) supplemented with 1 mg/L vitamin B12 (B12) (VWR, Darmstadt, Germany) or on BHI agar (Carl Roth) + 1 mg/L B12 at 37 °C for 48 h. Default parameters were used for all protocols and software unless otherwise specified. Genomic DNA was extracted from 4 mL of culture (48 h) using the ZR fungal/bacterial DNA miniprep kit (Zymo Research, Freiburg, Germany) Illumina sequencing and the Genomic Micro AX Bacteria+ kit (A&A Biotechnology, Gdynia, Poland) for Oxford Nanopore sequencing. DNA quality assurance and quantitation were performed using the Qubit fluorometer and dsDNA BR Assay kit (Life Technologies, US). Additional sequence data used in this study was shown in Table 3.

### 2.2. Genome Sequencing, Assembly and Annotation

#### 2.2.1. Illumina MiSeq Short-Read Sequencing and Assembly

The sequencing library was prepared with an Illumina TruSeq nano DNA LT library prep kit (Illumina, San Diego, USA) and sequenced on the Illumina MiSeq sequencer using the Illumina MiSeq V3 reagent kit according to the manufacturer’s protocols. The fastq files were quality controlled by FastQC 0.11.7 and assembled using SPADes Version 3.10.0 on the PATRIC BRC website (https://patricbrc.org/) [27]. The assemblies were imported into the commercial Geneious software (version 9.1.8) for sequence analysis (https://www.geneious.com). Contigs of less than 300 bp or with coverage <20× were removed manually and the remaining assemblies were exported as fasta files. Possible contamination of the assemblies was screened using the ContEst16S tool [28]. The genome quality was assessed with the PATRIC quality tool [27].

#### 2.2.2. MinION Nanopore Long Read Sequencing and Assembly

Long sequence reads were obtained by library preparation with the MinION 1D Native DNA barcoding genomic DNA protocol (with the Oxford Nanopore Ligation Sequencing Kit (EXP-NBD104) and the Oxford Nanopore Native Barcoding Expansion Kit (SQK-LSK 109)) and by sequencing using an Oxford Nanopore MinION MK1B sequencer with a MinION Flow Cell (R9.4.1) and a Flow Cell Priming Kit (EXP-FLP002). The long sequence reads of the *H. sporothermodurans* type strain DSM 10599^T^ were obtained using the same protocol, but with the following modifications, that were required due to unusual clumping of the magnetic beads during purification. For DNA clean-up of this strain, it was necessary to dilute the samples by 1:5 before adding the magnetic beads and to elute the DNA at 70 °C. Furthermore, this library was sequenced on a MinION MK1B sequencer but with a Flongle Flow Cell (FLO-FLG001). Reads were extracted from FAST5 files using the Guppy pipeline v. 2.3.1. (https://nanoporetech.com), demultiplexed by Porechop [21] and filtered by Nonofilt (v. 2.6.0) [29]. The reads were trimmed by the FastQC Utilities at PATRIC [30]. Trimmed MinION reads were hybrid assembled together with corresponding Illumina short-reads on the PATRIC website https://www.patricbrc.org/ [30] with Unicycler (no trim, 4 racon iterations, 0 pilon iterations, 300 bp minimum contig length and 30× min contig coverage). The assemblies were imported into Geneious^®^ and if more than 1 contig was generated by Unicycler, the smaller contigs were mapped against the largest contig and well-mapping (redundant) sequences were removed from the assembly. Additionally, the ends of the largest contigs were checked for overlap and circularized if they matched. Circular chromosomes were rotated to *dnak* as starting point. Possible contamination of the assemblies was screened using the ContEst16S tool [22]. The genome quality was assessed with the PATRIC quality tool [21].

#### 2.2.3. Annotation, Molecular Analysis and Core Gene Detection

All assembled contigs were annotated using the NCBI Prokaryotic Genome Annotation Pipeline version 4.10 [31,32] with default parameters. All annotated genomes are available with DDBJ/ENA/GenBank, the accession number given in Table 4. Molecular and general analysis of assemblies, annotations, coverage and genetic structure were investigated using Geneious^®^ 9.1.8. For generating the genome’s core and accessory genes, the assembled contigs were annotated with Prokka V1.14.5 [33] and the resulting gff3 files were processed with the pangenome pipeline Roary V3.11 [34]. Prokka and Roary were carried out with default settings at the Galaxy Europe V1.3 cloud (https://usegalaxy.eu/) [35]. Due to illogical results (see results), different settings of the pipeline were checked (minimum percentage identity for blastp, percentage of isolates a gene must be in to be core, splitting paralogs, maximum number of cluster) to find the cause for the observed phenomenon. For comparison, we increased the Protein Identity interval for blastp to 99% (95% default).

### 2.3. Overall Genome Related-Indexes and Phylogenomics-Based Subtyping

#### 2.3.1. 16S rRNA Gene Phylogeny

Pairwise sequence similarities were calculated using the 16S rRNA gene sequences available via the GGDC web server [36] at http://ggdc.dsmz.de/. Phylogenies were inferred by the TYGS web server [37]. The following steps were done by this service: The extraction of the 16S rDNA gene sequence from whole genome data, determination of the closest type strains by blast+ comparison and a multiple sequence alignment was created with MUSCLE [38]. Maximum likelihood (ML) and maximum parsimony (MP) trees were reconstructed from the alignments with RAxML [39] and TNT [40], respectively. For ML, rapid bootstrapping in conjunction with the autoMRE bootstopping criterion [41] and subsequent search for the best tree was used; for MP, 1000 bootstrapping replicates were used in conjunction with tree-bisection-and-reconnection branch swapping and ten random sequence addition replicates.

#### 2.3.2. Core Gene-Based Codon Tree

For further phylogenetic investigations a codon tree based on up to 1000 core genes was generated with PATRIC BRC [30]. The default was set for Codon Trees, which utilizes the amino acid and nucleotide sequences from up to 1000 PATRIC’s global Protein Families (PGFams). These are picked randomly to build an alignment and then generate a tree based on the differences within those selected sequences. Both the protein (amino acid) and gene (nucleotide) sequences are used for each of the selected genes from the PGFams. Protein sequences are aligned using MUSCLE [38], and the nucleotide coding gene sequences are aligned using the codon align function of BioPython [42]. A concatenated alignment of all proteins and nucleotides was converted to a phylip formatted file. Then a partitions file for RaxML [39] was generated, describing the alignment in terms of the proteins and then the first, second and third codon positions. Support values were generated using 100 rounds of the “Rapid” bootstrapping option of RaxML.

#### 2.3.3. Cohesive Genomic Sequence Comparisons (ANI, dDDH)

The overall genome-related indexes, i.e., average nucleotide identity (ANI) and digital DNA-DNA hybridization were used to compare the *H. sporothermodurans* group species. The Blast+ (ANIb) from all pairwise genome comparisons were computed at the online service JSpeciesWS [43]. Digital DNA-DNA hybridization (dDDH) values were estimated at GGDC (Genome-to-Genome Distance Calculator) using GGDC 2.1 with BLAST+ alignment and recommended formula 2 [36]. The proposed and generally accepted species boundary for ANI and dDDH values are 95–96% and 70%, respectively [44].

#### 2.3.4. Identifying Single-Nucleotide Polymorphisms (SNPs)

The online service *CSI Phylogeny-based subtyping* (SNP CSI Phylogeny 1.4) calls SNPs and was performed using whole genome sequence data (https://cge.cbs.dtu.dk/services/CSIPhylogeny/) [45]. The hybrid sequence assembly from *H. sporothermodurans* DSM 10599^T^ was used as reference genome and the raw reads (Illumina short-read sequence) were uploaded for each strain. Due to the unavailability of Sequence Read Archive (SRA) data (raw reads) for strain B4102 in the databases, the assembled data stored in the GenBank database were used. The analysis was submitted with the following parameters: a minimal depth at SNP positions of 25 reads, a minimal relative depth at SNP positions of 10%, a minimal distance between SNPs of 10 bp, a minimal Z-score of 1.96, a minimal SNP quality of 30, a minimal read mapping quality of 25 and ignore heterozygous SNPs activated. The pipeline and the parameters were described by Kaas et al. [45] and evaluated by Saltykova et al. [46]. Briefly, the pipeline performs the following steps: reads are mapped to the reference genome and SNPs are called. The SNPs are then filtered according to the submitted parameters in order to obtain a higher quality and positions that fail validation in at least one of the mapping steps are excluded from the SNP matrix.

## 3. Results

### 3.1. Whole Genome Sequencing (WGS)

In this study, we sequenced 29 *H. sporothermodurans* and two presumptive *H. vini* strains by short-read sequencing. The two presumptive *H. vini* strains (identification of the strains was based on previous 16S rRNA gene Sanger sequencing) could be reclassified as *H. sporothermodurans* (see below). Additionally, we sequenced the type strains *H. sporothermodurans* DSM 10599^T^, *H. vini* JCM 19841^T^ and *H. oleronia* DSM 9356^T^ using a combination of short- and long-read sequencing technologies. Sequencing and analysis was performed according to the methodology proposed by Chun et al. [44]. The quality of the raw reads was checked with FastQC, genome assembly quality was assessed with PATRIC BRC and contaminations were excluded by using the ContEst16S software. A summary of the sequenced genome characteristics is shown in Table 4. The data have been deposited to BioProject accession number PRJNA639094 in the NCBI BioProject database.

For the short-read-sequenced *H. sporothermodurans* genomes, sizes ranged from 3,372,204 bp to 3,886,000 bp with a GC content of 35.75 to 36.01 mol%. The number of gene coding sequences (CDS) ranged from 3758 to 4115. The type strain *H. sporothermodurans* DSM 10599^T^ assembly yielded 302 contigs after Illumina short-read sequencing, but could be improved to 1 contig by performing a hybrid assembly of Illumina short-read and MinION long read sequences. Using this method, the *H. oleronia* strain DSM 9356^T^ could be assembled into a single closed chromosome sequence, opposed to the short-read approach, which resulted in >500 contigs. When compared to the short-read assembly, the hybrid assembly noticeably increased the genome size from 3,839,826 bp to 4,417,946 bp (15.1%) for *H. sporothermodurans* DSM 10599^T^ and from 5,083,966 bp to 5,198,760 bp (2.3%) for *H. oleronia* DSM 9356^T^. Similarly, the GC content increased from 35.75 to 36.34 mol% GC for *H. sporothermodurans* DSM 10599^T^ and from 34.97 to 35.13 mol% GC for *H. oleronia* DSM 9356^T^. While the number of genes increased from 4032 to 4554 CDS for *H. sporothermodurans* DSM 10599^T^, a decrease from 5530 to 5397 CDS was observed for *H. oleronia* DSM 9356^T^. The massive increase in genome length, GC content and the number of genes in the hybrid assembly approach of *H. sporothermodurans* DSM 10599^T^ could be explained by an elevated detection of repetitive sequences like transposons, integrases and other mobile genetic elements, which varied between 1700 and 4000 nucleotides in size. In the short-read assembly (302 contigs) of *H. sporothermodurans* DSM 10599^T^ there were often residues of repeat regions located at the contig ends. In the corresponding hybrid assembly, these repeat regions were distributed over the whole genome sequence and are identified with a number of 340 regions (by Geneious^®^). This number of repeat regions is similar to the contig number in the short-read assembly. This finding suggests a correlation between the number of contigs in short-read assembly and the number of repetitive regions in the respective genome. In the short-read assembly, there are several potential matching short-reads at the regions of repetitive sequences (>500 bp), where the assembly has to be terminated and the end of a contig arises. With hybrid assembly (using long-reads of several kilobases in length), these repetitive regions could be continuous sequenced and assigned. This allowed the contigs to be placed in the correct order and additional genes were therefore found within the repetitive regions.

In addition, this study presents the first whole genome sequence of *H. vini* JCM 19841^T^. The sequence length, mol% GC content and gene number determined by the hybrid assembly for the *H. vini* JCM 19841^T^ genome were 4,309,805 bp, 35.97 mol% GC and 4373 CDS, respectively (Table 4).

Analysis of protein-encoding genes revealed that the genomes of the three characterized *Heyndrickxia* species (*H. sporothermodurans, H. oleronia* and *H. vini*) share 164 core genes (Appendix A). Surprisingly, *H. sporothermodurans* and *H. vini* share 965 core genes, while only 156 core genes were detected for *H. sporothermodurans* and *H. oleronia*. The latter number lies below the amount of core genes determined for all three species (n = 164) and thus suggests a suboptimal analysis (Roary pipeline) of the sequencing data as the numbers of core genes are expected to increase with a decreasing number of species examined. In order to reveal the underlying cause, the effects of slight changes to the parameters of the analysis pipeline were examined: When the protein identity interval was increased (sets the minimum percentage identity for protein match) to 99% (default was 95%), expectedly, less core genes were found, but the ratio of core genes between two species (44 genes) and all three species (42 genes) hinted towards an overall more reliable result (Appendix A). The observed anomaly (158 core genes for *H. sporothermodurans* + *H. oleronia* and 164 for *H. sporothermodurans* + *H. oleronia* + *H. vini*) was thus identified as a technical artifact in the blastp calculation (Roary pipeline). While the artifact had no significant influence on other results, it highlights the necessity of thorough manual examination of complex computational analyses.

Within the *H. sporothermodurans* species (n = 36 genome sequences) 1837 core genes (100% of strains harbor these genes), 490 soft core genes (95–99% of strains), 1923 shell genes (15–95% of strains), 2797 cloud genes (1–15% of strains) and 7047 different genes could be identified in total (1–100% of strains). Removing the more distant *H. sporothermodurans* strains SAD and B4102 (see below), the number of core genes within *H. sporothermodurans* genomes increased to 2187 (5649 different genes in total).

### 3.2. Heyndrickxia Sporothermodurans Interspecies Analysis

Phylogenetic tree based on 16S rRNA gene sequences of type strains of the *Heyndrickxia sporothermodurans* related species was done by the TYGS platform [37] and is shown in Figure 1. In the phylogenetic reconstruction *H. sporothermodurans*, *H. vini* and *H. oleronia* are related and cluster closely in a monophyletic group, which we termed the *H. sporothermodurans* group and in which *H. sporothermodurans* shows a closer relationship with *H. vini* than with *H. oleronia*.

Using the complete genome sequences from hybrid assemblies of the type strains of *H. sporothermodurans*, *H. vini* and *H. oleronia*, a phylogenomic tree based on core genes was constructed. In this phylogenomic analysis the *H. sporothermodurans* and *H. vini* type strains clustered well apart from the *H. oleronia* type strain (Figure 2), confirming that *H. sporothermodurans* is more closely related to *H. vini* than to *H. oleronia*. Based on a core genes analysis performed with 595 genes, *Margalitia (M.) shackletonii* and *M. camelliae* form a monophyletic clade. Phylogenetically more distant to *H. sporothermodurans,* and without distinct clades, are *W. ginsengihumi*, *B. acidicola*, *S. fordii* and *C. firmus*. In addition, *H. vini* formed a clearly separated branch in the more detailed maximum-likelihood phylogenomic tree (Figure 3) confirming previous conventional DNA-DNA hybridization results that *H. vini* is distinguishable from *H. sporothermodurans* at the species level. The phylogenomic tree furthermore shows that the strains M5572-6 isolated in Jordan and M5352-102 from Italy, that were previously identified as *H. vini* on the basis of 16S rRNA sequence analyses (sanger sequencing), clustered closely together with other *H. sporothermodurans* strains in the phylogenomic analysis, indicating that these strains were previously identified incorrectly and were in fact also *H. sporothermodurans* strains (Figure 3).

### 3.3. Heyndrickxia Sporothermodurans Intraspecies Analysis

In order to increase the resolution of the relationships of *H. sporothermodurans* strains, in a first step *H. vini* and then *H. sporothermodurans* strains showing less close relationship (SAD and B4102, see Figure 3) were omitted from the phylogenomic analyses. A core gene based phylogenomic tree for all available *H. sporothermodurans* strains, without *H. vini* type strain, is shown in Appendix A. A SNP analysis was also carried out (Appendix A) to confirm the results using a second approach. Both analyses showed that there was a general clonal relationship between all strains, with the exception of the strains SAD (from South Africa) and B4102 (from the Netherlands), which supports the observation from Figure 3. These two strains clustered relatively far away (Figure 3, Appendix A) and showed an excessive number of SNPs in relation to the type strain (1,548,865 positions were found in all analyzed genomes, 35.06% of reference genome), i.e., 7890 SNPs (strain SAD) and 4931 SNPs (strain B4102) versus 26 to 98 SNPs within the other 34 strains (type strain as reference, Appendix A).

To address the question of whether these two strains could nevertheless be unequivocally identified as *H. sporothermodurans,* or whether they constituted a different species, digital whole genome comparisons were performed. Digital whole genome comparisons based on average nucleotide identities (ANIs) or genome-genome-distance calculations (GGDCs) are considered as gold standard for species taxonomy [44]. Generally, a GGDH value >70% and/or an ANI >95–96% of the analyzed strain indicates that it belongs to the matching type strain [44]. The values of 84.50–90.20% GGDC and 97.70–98.38% ANIb showed that SAD and B4102 belong to the species *H. sporothermodurans* (Appendix A). However, a clear genetic distance from the type strain was noted when compared to all other strains, with values of 99.5–99.9% GGDH and 99.75–99.94% ANIb (Appendix A).

Due to these differences, the strains SAD and B4102 were not assigned to the phylogenetic “*type strain group*”, which consists of the majority of the *H. sporothermodurans* isolates and the type strain, which cluster together closely (Figure 3). A further phylogenetic analysis based on the core genes and accessory genes was done for this *type strain group* alone, without the less-related strains SAD and B4102, in order to obtain a more detailed view on the internal relationships of these strains (Figure 4).

Overall, the *type strain group* has only a few SNPs (<100) and is genetically very homogeneous. Nevertheless, when looking within the *type strain group* at high resolution, there was also considerable heterogeneity between strains, especially evident from the strain L1_142 and L1_141 (Germany), BR3 (Brazil) and the single isolates of Ecuador, Iran and Jordan, which formed individual lineages (Figure 4).

Within the *type strain group* (Figure 4) there appeared to be a definite clonal relationship between strains of similar geographical origin. Thus, strains from South Africa clustered closely together, as did most strains from Germany, Italy and Czech Republic. Similarly, isolates from the same dairy at different timepoints (dairy C: L1_149, L1_150 (yellow) and dairy E: L1_158, L1_159, L1_160, L1_161 (green, Figure 4) clustered closely, together indicating a close phylogenomic relationship. Also, for isolates in dairy D (red, Figure 4) the strains grouped closely and showed a high degree of relatedness, especially when the times of isolation were close as it was the case for e.g., isolate L1_151 (isolation date 23 August 2006), L1_152/L1_153 (19 July 2007), L1_154 (17 July 2007) and L1_155 (31 July 2007). However, later isolates showed a significantly larger genetic difference (L1_156, L1_157) and cluster apart. Similarly, the blue marked isolates L1_143, L1_144, L1_145 from dairy B (Figure 4) which were isolated in a similar time period (18 June 2003, 19 August 2003, 26 August 2003) also formed a clonal lineage, which was separated from the 2 other strains L1_141 (23 June 1999) and strain L1_146 (02 December 2003), which were isolated at an earlier and later period from this dairy.

The differences in single-nucleotide polymorphisms (SNPs) between the members of a closely related clade e.g., L1_152, L1_153, L1_154 and L1_155 (red) are 3–7 SNPs (Figure 4 and Appendix A). In comparison, the closely related strain L1_151 (red) from the same dairy showed 13 to 17 SNPs to this group. Other highly clonal related groups exhibited 4 to 8 SNPs to each other (dairy E: L1_158, L1_159, L1_160, L1_161, green in Figure 4), 5 to 12 SNPs (dairy B: L1_143, L1_144, L1_145, blue in Figure 4) and 6 SNPs (dairy C, L1_149, L1_150, yellow in Figure 4).

In order to assess the closely related clade even more precisely, a core genome calculation within the clonally closest related isolates was performed and accessory genes were compared. The isolates L1_152, L1_153, L1_154 and L1_155 of dairy D (red) contain 3747 genes in total, of which 3418 are core genes (100% of isolates harbor these genes) and 329 are accessory (shell) genes (15–95% of isolates harbor these genes). The isolates L1_158, L1_159, L1_160 and L1_161 of dairy E (green) contain 3753 genes in total, of which 3288 are core genes and 465 are accessory genes. The isolates L1_143, L1_144 and L1_145 of dairy B (blue) contain 3969 genes in total, of which 3679 are core genes and 290 are accessory genes (shell) genes.

## 4. Discussion

High-throughput sequencing technology development has led to a significant improvement for the discrimination of bacteria at the genomic level. This study has extended the genome sequence data from 5 so far examined to 36 *H. sporothermodurans* strains, and furthermore improved the existing *H. sporothermodurans* (DSM 10599^T^) and *H. oleronia* (DSM 9356^T^) type strain genome sequences. In addition, this study is the first to present the complete genome of *H. vini* (JCM 19841^T^).

The here presented data show that a combination of short- and long-read sequencing data results in a considerable decrease of contig numbers from a few hundred to generally a single complete chromosome and thus represents a superior genome assembly approach. Using this approach, we frequently noticed an increase in genome length compared to the short-read assemblies. For *H. sporothermodurans*, a genome extension of 15% was achieved by long-read sequencing of the overlapping areas between the short-read contigs. At the ends of short-read contigs and in the overlapping areas, repeat regions (mobile elements, e.g., IS1182 family transposase) were frequently detected (data not shown). For species with many repetitive elements, long read sequencing is necessary to obtain the complete genome information, because repeats lead to unresolvable loops in the assembly graph that are ultimately fragmented into contigs [47,48]. It may be speculated that these repeat sequence areas are the reason for the unusually high number of contigs obtained by short-read sequencing, as it is not possible to assemble repeat regions with short-reads that occur multiple times, because short-reads alone cannot resolve repetitive genomic regions that are longer than their read length [47]. As a consequence, the numbers of small contigs, which were generated based on short-read sequencing would correlate to the number of repetitive elements present in the genome of the sequenced bacterium.

The reason for the high abundance of repeat regions in the genome of *H. sporothermodurans* has yet to be uncovered. However, it has been demonstrated that such regions can increase resistance of bacterial spores towards environmental stresses like heat exposure [49] and we hypothesize that similar mechanisms would explain the occurrence of repeat elements in the *Heyndrickxia* species presented in this study.

A striking feature of the *H. sporothermodurans* genome is the high degree of relationship in the respective core genes of strains, in contrast to the diversity of the respective accessory genes. Apparently, the essential core genes are much less affected by horizontal gene transfer (HGT) than the rest of the genome. In evolutionary terms, *de novo* mutations in *Bacillus* are probably less important than HGT for the variability of genomes [50,51]. For *B. subtilis*, it has been shown that after 504 generations up to 2% of the genome has been replaced by HGT [51]. Therefore, foreign DNA from related species was taken up and transferred or replaced by homologous recombination in integration hotspots. In our study we did not detect plasmids in the genomes, so they are less common in *H. sporothermodurans* and probably play a minor role in HGT. However, repeat regions that we found in *H. sporothermodurans* represent a general mechanism for horizontal gene transfer [50,51,52]. Core genes are not so strongly affected by this influence and remain largely stable, so that core genome and SNP analyses show a very high degree of homogeneity, although massive changes in the rest of the genome have already occurred. This would explain the very close relationship within the type strain group on the one hand; the less-related strains SAD and B4102, on the other hand, would be classified as additional clonal related lines, each forming a separate phylogenetic clade.

Two *H. vini* strains (previously identified on the basis of Sanger sequenced 16S rDNA analysis) could be identified as *H. sporothermodurans* on the basis of whole genome sequencing results. This clearly demonstrates that traditional 16S rRNA gene analysis in this case does not have sufficient resolution to discriminate between *H. vini* and *H. sporothermodurans*. The reason is probably the heterogeneity of the 16S rRNA gene copies of *H. sporothermodurans*, where some variants show a higher similarity to the published *H. vini* 16S rDNA (data not shown) than others. *H. sporothermodurans* and *H. vini* are indeed the most closely related, shown on the basis of 16S rDNA, core genes and digital whole genome comparisons (88.97% ANIb and 39.70% GGDH). Other species closely related to this *H. sporothermodurans*/*vini* clade are *H. oleronia, M. shackletonii* and *M. camelliae*. Interestingly, based on an analysis of core genes performed with 595 genes, *M. shackletonii* and *M. camelliae* form a monophyletic clade. Less related to *H. sporothermodurans* are *Weizmannia (W.) ginsengihumi*, *B. acidicola, Siminovitchia (S.) fordii* and *Cytobacillus firmus*, which deviates from previously reported work [4]. Digital DNA-DNA hybridization values between *H. sporothermodurans* and related strains generally support the traditional wet-lab data (Table 5) for speciation, but we found no correlation in the degree of inter-species relationships [53] in this group of bacteria. For example, DNA-DNA hybridization indicates a higher relationship between *M. shackletonii* and *H. sporothermodurans* than between *H. oleronia* and *H. sporothermodurans*, but the core genome analyses (595 genes) clearly shows a greater relationship between *H. oleronia* and *H. sporothermodurans*. As a result, DNA-DNA hybridization methods were not suitable for the determination of the degree of inter-species-relationships with values below 70%.

The new classification of *Bacillus* species was published by Gupta et al. in 2020, which proposes a comprehensive reclassification and renaming of many *Bacillus* species [5]. Based on genomic comparisons, 17 distinct *Bacillus* species clades were identified [5]. A monophyletic clade was postulated for *B. oleronius* and *B. sporothermodurans* (Oleronius Clade), for which the genus *Heyndrickxia* gen. nov. was proposed [5]. Accordingly, *B. oleronius* was proposed to be renamed to *Heyndrickxia oleronia* and *B. sporothermodurans* to *Heyndrickxia sporothermodurans* [5]. Further clades were also proposed for *B. shackletonii* and *B. camelliae* (Shackletonii Clade, *Margalitia* gen. nov.), *B. ginsengihumi* (Coagulans Clade, *Weizmannia* gen. nov.) and *B. fordii* (Fordii Clade, *Siminovitchia* gen. nov.), among others [5]. The renaming of *B. firmus* into *Cytobacillus firmus* has also been proposed [6]. Our core genome-based analyses support the proposed monophyletic clades Oleronius and Shackletonii in general (see Figure 2). However, our data show two distinct subclades within the Oleronius clade. One subclade consists of *H. sporothermodurans* and *H. vini*, the other subclade of *H. oleronia*. The shared core genes from the Roary pipeline (964 core genes *H. sporothermodurans* -*H. vini* vs. 158 core genes *H. sporothermodurans* -*H. oleronia*) confirm the different species subclades. It is therefore important to include genome data such as we presented here in future taxonomical *Bacillus* analysis (especially *H. vini)*. As the reclassifications were already valid (http://lpsn.dsmz.de as reference) at the time of writing, we used the new, valid species/taxon names in this study. Although the new names may initially cause some confusion, the proposed developments are justified and improve the taxonomic classification of many *Bacillus* species.

The intraspecies analysis of all available *H. sporothermodurans* genomes revealed that 34 of 36 strains belonged to the same ecotype [55], which was termed “*type strain group”* accordingly in this study. The bases for these groupings were the high similarities between the strains to the type strain DSM 10599^T^ in both GGDH (>99.5%) and ANIb (>99.75%), as well as the supporting core gene homologies and single-nucleotide polymorphisms (SNPs) analyses. The strains SAD and B4102 do not belong to this *type strain group* (2 of 36 strains) due to the results of whole-genome-comparisons, core gene and SNP analyses (e.g., GGDH < 90.2% and ANIb < 98.38%). Heyndrickx et al. [4] described a higher heterogeneity for *H. sporothermodurans* strains based on protein levels (SDS-page subgrouping) and wet lab DNA-DNA hybridization (76–88%). There, the authors analyzed *H. sporothermodurans* strains from different origins (UHT-milk, raw milk, soy, feed concentrate, pulp, silage), while only UHT-milk isolates were used in the present study. The limited strain variation detected here indicates that UHT-processing treatment of raw milk results in the selection of a specific ecotype.

It is difficult to provide a satisfactory answer to the question of whether and how long a strain may persist in a single dairy, or whether there are new contaminating strains being continually brought into the dairy. It can, however, be speculated that isolate L1_151 persisted in dairy D and developed into the clonal related strains L1_152, L1_153, L1_154, L1_155 (3–7 SNPs) within the 11 months these strains were isolated. The SNPs between isolate L1_151 (origin) and L1_152, L1_153, L1_154, L1_155 were 18, 16, 21, 22 SNPs, respectively. However, this does not correspond with the times of sampling, if accumulation of SNPs during time would be assumed. Additionally, the clonal isolates L1_152, L1_153, L1_154 and L1_155 harbor 3418 core and 329 accessory genes. This was quite a surprising high rate of accessory genes for strains with such low numbers of 3–7 SNPs in their core genomes. One explanation would be the coexistence and coevolution of different *H. sporothermodurans* strains in one dairy, even if they descended from a common ancestor. This coevolution of different strains can lead to the accumulation of SNPs at different speeds over time [56]. Furthermore, it has been shown by Dettling et al. that spore formers can persist for up to 24 months in milk powder processing lines [56].

The strains from Germany also showed individual clusters in which strains showed clonal relationships, but different clusters of German strains also showed that these probably presented individual lineages (Figure 3). A remarkable observation from a previous study done with a limited set of isolates showed that one clone, termed HRS, was involved in the majority of the cases of UHT milk spoilage [14]. Indeed, when looking at individual clusters in this study up close, clonal relationship could also be determined between strains, often when these were isolated in the same or in consecutive years (Figure 4). There seems to be not just one dominant clone in a specific dairy processing plant, but a parallel development from an original ancestor. As an example, isolates L1_158, L1_159, L1_160, L1_161 from a single dairy show differences from 4 to 8 SNPs (between the strains), although all were isolated on the same day. On the other hand, there are 4 months between the sampling of isolates L1_149, L1_150, which differ only by 6 SNPs (Appendix A).

In general, there is no strict rule for a separation between the terms clone and strain. For *Listeria* (*L*.) *monocytogenes* outbreak analysis, the distances are set at 4 to 25 (up to 63) SNPs for a clonal line, depending on the species and outbreak period (isolation time spanned several months or years) [52]. Other authors concluded that *L. monocytogenes* strains with under 10 SNPs can occur in different retail deli environments without any links of known transmission [57]. Today, direct clones should have almost no SNPs, since the sequencing errors are very small [23].

Only when horizontal gene transfer and the core genome are evaluated together, is it possible to distinguish between strain and clonal line [52]. Here, we can assume for the isolates with <10 SNPs difference in their genomes are classified as clonal lines derived from the same strain. More than 10 SNPs indicates a close relationship but corresponding isolates likely belong to a different strain. In this sense, the isolates L1_158, L1_159, L1_160, L1_161 (4 to 8 SNPs) belong to one strain, also L1_149, L1_150 (6 SNPs), L1_156, L1_157 (4 SNPs) and L1_152, L1_153, L1_154, L1_155 (3–7 SNPs) and so on. However, this is a very arbitrary classification, whose scientific basis is highly debatable and which is only mentioned here for the sake of completeness.

However, this very close relationship indicates, that due to the high thermotolerance of their spores, bacteria persist within a milk processing environment for long time periods of up to several months. Moreover, the occurrence of phylogenetic clusters often correlated with their respective geographic origin, indicating that these heat-resistant bacteria not only co-exist for extended periods of time in specific milk processing environments, but also display a similar dissemination behavior between closely located processing environments. Supporting this assumption, we determined one monophyletic cluster containing highly related strains, which originated from proximal geographic sites, i.e., Italy, Albania and Czech Republic.

It is unclear which physiological adaptations (heat resistance, spore production, biofilm formation, etc.) were triggered by the genetic changes. For this, further studies should be carried out, if possible with a greater heterogeneity of strains. Determining the heat resistance of the spores is one of the important points to clarify if some clusters are more heat resistant than others. It is possible that isolates from one dairy may become more resistant over time. These results on heat resistance should then be compared with the genomic and genomic change (i.e., evolutionary) data, similar to those presented in this study. Also, the heat resistance of *H. vini* spores has not been estimated so far, this should also be done in future investigations. Thermal preservation is of great importance for the food industry. If the thermal processes are not sufficient for spore-inactivation, this can lead to premature food spoilage. Our investigations show the persistence of *H. sporothermodurans* for months in food processing plants. For this reason, molecular detection systems should also be developed for selective and sensitive detection of *H. sporothermodurans* and other relevant *Bacillus* species contamination to prevent financial losses for companies from loss of quality of the processed milk products.

## 5. Conclusions

This study presents the complete genomes of the type strains *H. sporothermodurans* DSM 10599^T^, *H. oleronia* DSM 9356^T^ and *H. vini* JCM 19841^T^ as well as the genomes of 31 *H. sporothermodurans* strains isolated from UHT-treated milk from different continents. This study is the first to present the complete genome of *H. vini* (JCM 19841^T^). The type strains (*H. sporothermodurans*, *H. oleronia* and *H. vini*) were clustered into one phylogenetic clade, whereby *H. sporothermodurans* was more closely related to *H. vini* than to *H. oleronia*. Moreover, this clade was found to be distinct from the *M. shackletonii* and *M. camelliae* cluster.

Previous work reported difficulties in assembling *Heyndrickxia* species genomes using a short-read sequencing approach [13,18,20]. Therefore, we utilized hybrid assembly of Illumina short-read and MinION long-read sequences of type strain *H. sporothermodurans* DSM 10599*^T^*, which mostly yielded a single coherent chromosome sequence instead of a few hundred contigs. Furthermore, genome size and G + C value differed significantly between both assembly approaches. We could detect a major increase in genome length and a minor increase in the mol% G + C content. The high amount of contigs obtained after short-read sequencing is probably caused by the occurrence of mobile elements which are overrepresented in *H. sporothermodurans*. As the assembly of genomes containing multiple repetitive genetic elements can be challenging, this species can be considered a model organism for improving and evaluating sequencing methods and assemblies.

Furthermore, a high clonal relationship of core genes was shown in 34 of 36 *H. sporothermodurans* strains, which were classified as *type strain group* accordingly. The remaining two strains (SAD and B4102) could be identified as *H. sporothermodurans* as well, based on digital whole genome comparison. Nevertheless, both strains probably belong to distinct phylogenetic clades due to their clear genetic distance from the *type strain group*.

In conclusion, our results indicate that clonally related isolates with <10 SNPs occur in the course of few months in industrial dairy environments and can persist and (co)evolve quickly. Therefore, this study represents the basis for the development of detection systems and avoidance strategies to prevent contamination of food with highly heat-resistant spore formers like *H. sporothermodurans*.

## Figures and Tables

**Figure 1 microorganisms-09-00246-f001:**
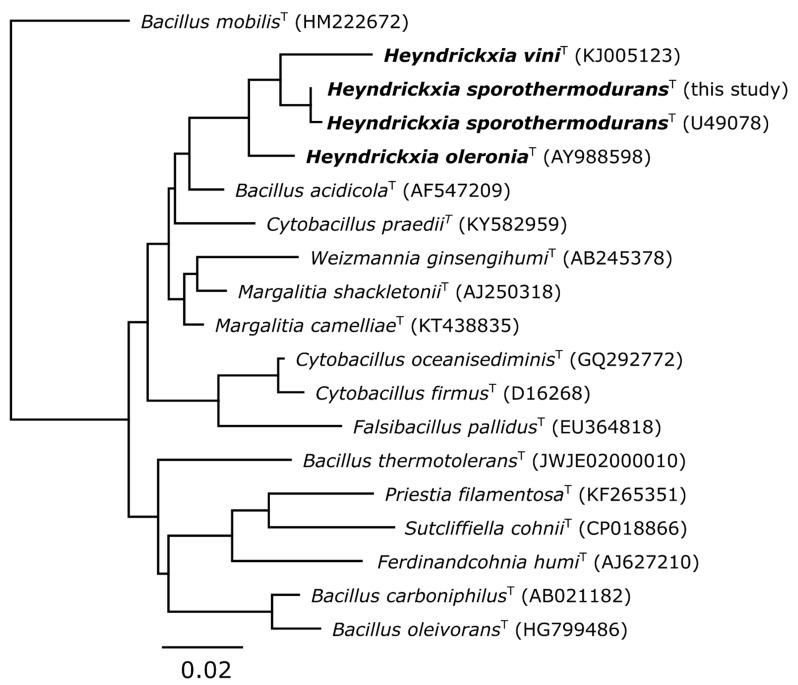
Phylogenetic 16S rDNA tree. The nearest type strains were selected, based on a blast+ 16S rDNA comparison. The phylogenic Maximum Likelihood (ML) tree inferred under the GTR + GAMMA model and rooted to *Bacillus mobilis*. The branches are scaled in terms of the expected number of substitutions per site.

**Figure 2 microorganisms-09-00246-f002:**
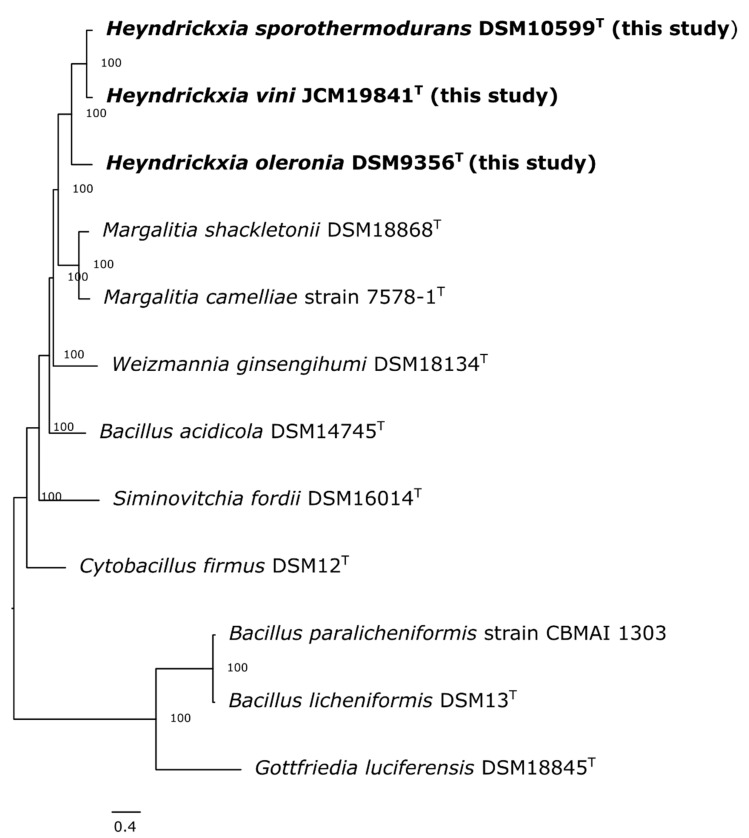
Phylogenetic tree analysis (Codon Tree method selects 595 single-copy genes (PATRIC PGFams) and analyzes aligned proteins and coding DNA using the program RAxML version 8.2.11). Species were selected based on 16S rDNA analysis (Figure 1) and literature reference [2,3,4]. *B*. *licheniformes*, *B*. *paralicheniformes* and *Gottfriedia luciferensis* were selected as outgroup. The branches are scaled in terms of the expected number of substitutions per site.

**Figure 3 microorganisms-09-00246-f003:**
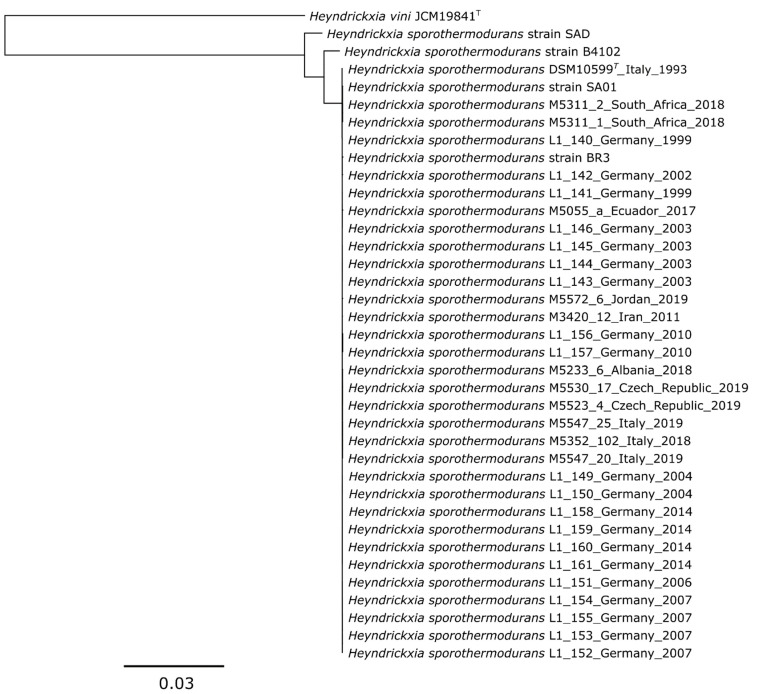
Phylogenomic tree analysis of *H. vini* and *H. sporothermodurans* strains (Codon Tree method which selects 1000 single-copy genes (PATRIC PGFams) and analyzes aligned proteins and coding DNA using the program RAxML version 8.2.11). The strains were named with the strain number, the country of origin and the year of isolation. Already published strains are DSM 10599^T^, SAD, B4102, SA01 and BR3 (see material section). The branches are scaled in terms of the expected number of substitutions per site.

**Figure 4 microorganisms-09-00246-f004:**
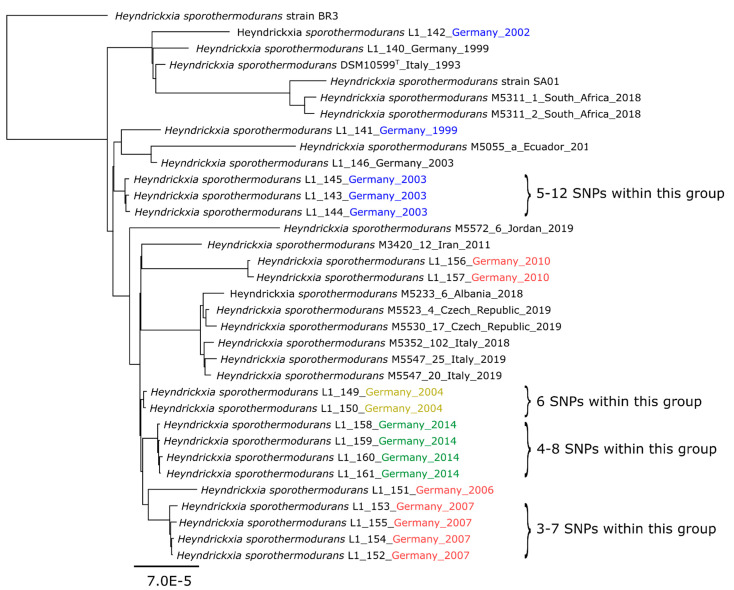
Phylogenomic tree analysis of the *H. sporothermodurans* type strain group, n = 34 (Codon Tree method selects 1000 single-copy genes (PATRIC PGFams) and analyzes aligned proteins and coding DNA using the program RAxML version 8.2.11, tree was rooted to strain BR3). Isolates from the same dairy were marked in color. The branches are scaled in terms of the expected number of substitutions per site.

**Table 1 microorganisms-09-00246-t001:** Previous (basonym) and new combinations (comb. nov.) of *Bacillus* species used in this study based on their reclassification as described by Gupta et al. (2020) [5,6].

Previous Combination (Basonym)	Publication	New Combination (comb. nov.)	Publication
*Bacillus sporothermodurans*	Pettersson et al., 1996	*Heyndrickxia* *sporothermodurans*	Gupta et al., 2020
*Bacillus vini*	Ma et al., 2017	*Heyndrickxia vini*	Gupta et al., 2020
*Bacillus oleronius*	Kuhnigk et al., 1996	*Heyndrickxia oleronia*	Gupta et al., 2020
*Bacillus shackletonii*	Logan et al., 2004	*Margalitia shackletonii*	Gupta et al., 2020
*Bacillus camelliae*	Niu et al., 2018	*Margalitia camelliae*	Gupta et al., 2020
*Bacillus ginsengihumi*	Ten et al., 2007	*Weizmannia ginsengihumi*	Gupta et al., 2020
*Bacillus fordii*	Scheldeman et al., 2004	*Siminovitchia fordii*	Gupta et al., 2020
*Bacillus firmus*	Bredemann and Werner 1933	*Cytobacillus firmus*	Patel and Gupta 2020
*Bacillus luciferensis*	Logan et al., 2002	*Gottfriedia luciferensis*	Gupta et al., 2020

**Table 2 microorganisms-09-00246-t002:** Strains used in this study. * DSMZ = German Collection of Microorganisms and Cell Cultures; TU Munich = Technical University of Munich; JCM Riken BRC = Japan Collection of Microorganisms.

Species	Strain	Isolation Country	Isolation Source	Isolation Date (and Dairy)	Provided by *
*H. sporothermodurans*	DSM 10599^T^	Italy	UHT milk	1993	DSMZ
	M3420-12	Iran	UHT milk	2011	Tetra Holdings GmbH
	M5055-A	Ecuador	UHT milk	2017	Tetra Holdings GmbH
	M5233-6	Albania	UHT milk	2018	Tetra Holdings GmbH
	M5311-1	South Africa	UHT milk	2018	Tetra Holdings GmbH
	M5311-2	South Africa	UHT milk	2018	Tetra Holdings GmbH
	M5523-4	Czech Republic	UHT milk	2019	Tetra Holdings GmbH
	M5530-17	Czech Republic	UHT milk	2019	Tetra Holdings GmbH
	M5547-20	Italy	UHT milk	2019	Tetra Holdings GmbH
	M5547-25	Italy	UHT milk	2019	Tetra Holdings GmbH
	L1_140	Germany	UHT milk	07 July 1999 A	TU Munich
	L1_141	Germany	UHT milk	23 June 1999 B	TU Munich
	L1_142	Germany	UHT milk	29 July 2002 B	TU Munich
	L1_143	Germany	UHT milk	18 June 2003 B	TU Munich
	L1_144	Germany	UHT milk	19 August 2003 B	TU Munich
	L1_145	Germany	UHT milk	26 August 2003 B	TU Munich
	L1_146	Germany	UHT milk	02 December 2003 B	TU Munich
	L1_149	Germany	UHT milk	22 June 2004 C	TU Munich
	L1_150	Germany	UHT milk	15 October 2004 C	TU Munich
	L1_151	Germany	UHT milk	23 August 2006 D	TU Munich
	L1_152	Germany	UHT milk	19 July 2007 D	TU Munich
	L1_153	Germany	UHT milk	19 July 2007 D	TU Munich
	L1_154	Germany	UHT milk	17 July 2007 D	TU Munich
	L1_155	Germany	UHT milk	31 July 2007 D	TU Munich
	L1_156	Germany	UHT milk	20 January 2010 D	TU Munich
	L1_157	Germany	UHT milk	20 January 2010 D	TU Munich
	L1_158	Germany	UHT milk	26 September 2014 E	TU Munich
	L1_159	Germany	UHT milk	26 September 2014 E	TU Munich
	L1_160	Germany	UHT milk	26 September 2014 E	TU Munich
	L1_161	Germany	UHT milk	26 September 2014 E	TU Munich
*H. vini* (based on 16S rRNA)	M5352-102	Italy	UHT milk	2018	Tetra Holdings GmbH
	M5572-6	Jordan	UHT milk (banana)	2019	Tetra Holdings GmbH
*H. vini*	LAM0415^T^/JCM 19841^T^	China	alcohol fermentation pit mud	before 2016	JCM Riken BRC
*H. oleronia*	DSM 9356^T^	France	termite	1996	DSMZ

**Table 3 microorganisms-09-00246-t003:** Additional sequence data used in this study.

Species	Strain	Isolation Country and Year	Isolation Source	GenBank	SRA	Publication
*H. sporothermo- durans*	BR3 (BR12)	Brazil, 2012	UHT milk	NAZC01000000	SRR8741693	[22]
	SAD	South Africa, 2015	UHT milk	NAZB01000000	SRR8732968	[22]
	SA01	South Africa, 2015	UHT milk	NAZA01000000	SRR8732969	[22]
	B4102	The Netherlands, 2012	Indian curry	LQYN00000000	N/A	[15]

**Table 4 microorganisms-09-00246-t004:** Characteristics of the sequenced genomes (BioProject PRJNA639094).

Strain	DDBJ/ENA/ GenBank Accession Number	Number of Raw Reads	Coverage	Genome Size	Mol GC%	Contigs	N50	Genes (Total)	tRNA	rRNA	SRA Accession no.
	***Heyndrickxia sporothermodurans (*** **basonym: *Bacillus sporothermodurans)***										
DSM 10599^T^ short-reads	JAEKDY000000000	1,662,070	104.64×	3,839,826	35.75	302	27,181	4032	96	33	SRR13249535
DSM 10599^T^ Hybrid assembly	CP066701	1,662,070 (Illumina) 232,544 (Nanopore)	191.36×	4,417,946	36.34	1	1 contig	4554	99	30	SRR13249535 (Illumina) SRR13347551 (Nanopore)
M3420-12	JABWUU000000000	1,214,420	82.24×	3,580,829	35.93	291	25,466	3713	77	29	SRR13249560
M5055-A	JABWUT000000000	1,160,538	77.17×	3,662,309	36.01	348	20,956	3890	97	35	SRR13249559
M5233-6	JABWUS000000000	1,167,612	76.00×	3,746,330	35.91	363	20,329	3910	79	30	SRR13249548
M5311-1	JABWUR000000000	1,211,006	81.10×	3,615,469	35.85	346	20,889	3819	78	32	SRR13249537
M5311-2	JABWUQ000000000	1,205,508	82.18×	3,550,370	35.83	333	20,966	3703	95	33	SRR13249530
M5523-4	JABWUP000000000	1,192,692	81.50×	3,569,406	35.95	341	20,150	3719	81	31	SRR13249529
M5530-17	JABWUO000000000	1,196,538	81.52×	3,576,306	35.94	338	20,329	3715	78	31	SRR13249528
M5547-20	JABWUN000000000	1,258,834	81.58×	3,764,103	35.93	372	20,038	4025	97	33	SRR13249527
M5547-25	JABWUM000000000	1,551,052	106.16×	3,534,820	35.91	342	20,854	3699	79	32	SRR13249526
M5352-102 *	JABWTR000000000	1,673,766	120.90×	3,376,234	35.88	332	20,219	3551	79	30	SRR13249538
M5572-6 *	JABWTQ000000000	1,661,774	111.93×	3,612,108	35.97	321	22,597	3780	97	33	SRR13249536
L1_140	JABWUJ000000000	1,300,432	82.43×	3,817,614	35.79	310	30,062	4026	96	33	SRR13249525
L1_141	JABWUI000000000	1,281,714	84.40×	3,684,244	35.79	319	24,263	3863	95	33	SRR13249558
L1_142	JABWUH000000000	1,265,780	82.99×	3,677,671	35.75	326	28,594	3886	79	32	SRR13249557
L1_143	JABWUG000000000	1,361,164	89.88×	3,676,925	35.84	315	25,507	3896	95	34	SRR13249556
L1_144	JABWUF000000000	1,446,376	90.58×	3,857,917	35.81	339	25,432	4115	96	35	SRR13249555
L1_145	JABWUE000000000	1,142,112	72.70×	3,811,445	35.83	326	25,433	4014	96	33	SRR13249554
L1_146	JABWUD000000000	1,476,620	93.85×	3,826,729	35.82	351	24,470	4042	96	33	SRR13249553
L1_149	JABWUC000000000	1,468,760	92.28×	3,886,000	35.84	336	23,762	4069	95	34	SRR13249552
L1_150	JABWUB000000000	1,232,426	80.78×	3,698,569	35.86	317	25,508	3850	95	34	SRR13249551
L1_151	JABWUA000000000	1,390,644	93.45×	3,628,299	35.85	320	26,528	3783	78	30	SRR13249550
L1_152	JABWTZ000000000	1,306,056	87.84×	3,605,120	35.87	313	26,883	3748	78	32	SRR13249549
L1_153	JABWTY000000000	1,389,634	95.93×	3,512,664	35.86	309	26,406	3694	78	32	SRR13249547
L1_154	JABWTX000000000	1,185,268	84.22×	3,418,001	35.85	301	26,933	3548	77	32	SRR13249546
L1_155	JABWTW000000000	1,832,218	121.31×	3,676,944	35.87	324	25,595	3843	96	33	SRR13249545
L1_156	JABWUL000000000	1,940,212	121.72×	3,877,835	35.90	342	24,263	4023	97	33	SRR13249544
L1_157	JABWUK000000000	1,770,320	119.84×	3,613,127	35.88	312	25,846	3751	78	31	SRR13249543
L1_158	JABWTV000000000	1,761,290	121.53×	3,535,146	35.84	323	23,762	3687	77	32	SRR13249542
L1_159	JABWTU000000000	1,724,638	124.93×	3,372,204	35.89	304	23,967	3484	77	32	SRR13249541
L1_160	JABWTT000000000	1,164,458	78.10×	3,633,431	35.86	334	24,295	3754	80	34	SRR13249540
L1_161	JABWTS000000000	1,814,922	120.51×	3,680,123	35.89	342	22,342	3785	78	31	SRR13249539
* formerly *H. vini* (based on 16S)											
	***Heyndrickxia*** ***vini (*** **basonym: *Bacillus vini)***										
LAM0415^T^/ JCM 19841^T^ Hybrid assembly	CP065425	2,323,916 (Illumina) 145,000 (Nanopore)	379×	4,309,805	35.97	1	circular	4263	97	30	SRR13249532 (Illumina) SRR13249531 (Nanopore)
	***Heyndrickxia oleronia (*** **basonym: *Bacillus oleronius)***										
DSM 9356^T^ Hybrid assembly	CP065424	1,700,898 (Illumina) 109,411 (Nanopore)	282×	5,198,760	35.13	1	circular	5186	145	36	SRR13249534 (Illumina) SRR13249533 (Nanopore)

**Table 5 microorganisms-09-00246-t005:** Comparison between traditional and digital DNA-DNA hybridization for *Heyndrickxia sporothermodurans* and closely related species.

	Traditional DNA-DNA Hybridization	Reference	Digital DNA-DNA Hybridization (This Study)
	Reference: *Heyndrickxia sporothermodurans* DSM 10599^T^		
*H. vini* JCM 19841^T^	33.3	[3]	39.70
*H. oleronia* DSM 9356^T^	16.0	[18]	22.10
*Margalitia shackletonii* DSM 18868^T^	42.0	[4]	23.8
*B. acidicola* DSM 14745^T^	35.0	[54]	26.1

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
