# Peer review of "Taxonomic Evaluation of the Heyndrickxia (Basonym Bacillus) sporothermodurans Group (H. sporothermodurans, H. vini, H. oleronia) Based on Whole Genome Sequences"

_microorganisms, 2021, doi:10.3390/microorganisms9020246_

Round 1

Reviewer 1 Report

This is an interesting paper describing the taxonomic relationships between three species of the B. sporothermodurans group on the basis of hybrid genome sequence analysis. The close relationship between B. sporothermodurans and B. vini was shown as well as between most of the strains of B. sporothermodurans. Nevertheless, by using SNP analysis, several clusters could be identified within B. sporothermodurans which correlated to a large extent with origin. This confirms earlier observations of clonal expansion of this highly heat resistant species in the dairy industry, but with a higher resolution on genome scale, more detailed relationships between individual strains could be established which leads to the interesting speculation of parallel evolution of different strains originating from a common ancestor. Furthermore, a large proportion of repeat regions associated with mobile elements were found which might be related to physiological traits of this organism causing its persistence in food processing environments.

There are some points to consider for improvement of the manuscript:

-an important remark is about the recent reclassification of members of this group to the new genus Heyndrickxia. Likewise, some other Bacillus species are used in the study which are also reclassified such as B. firmus to Cytobacillus, B. shackletonii to Margalitia, etc. The authors argue not  to use this new names and only use valid species names because at the moment of writing, these reclassifications would have not yet been completed. However, even at the moment of writing, these new names were already valid and should therefore preferably be used in the text and trees to reflect the correct taxonomic classification of Bacillus species. The authors can use the website http://lpsn.dsmz.de   to check the correct names of all species included.

-another important remark is that the authors included the not-validated species “B. mediterraneensis” (to be written between quotation marks) in the 16S rDNA tree, but not “B. obstructivus” which also belongs to the B. sporothermodurans clade according to Gupta et al. (2020). Furthermore, a genome is available for the latter not-validated species in NCBI database. It would be very valuable if this species (although not validated) could be included in (at least some) of the analyses.

-in the abstract, it should be described that 2 B. vini [sic] isolates have been reclassified as B. sporothermodurans [sic]. In Results, this reclassification should be described as soon as possible.

-Discussion is rather long and contains repetitions with Results, so Discussion can be reduced.

-Conclusion is rather long and can be condensed without giving specific details.

-L41: sometimes pink coloration of milk is observed as spoilage

-L75-76: 31 additional strains of B. sporothermodurans is confusing here in relation to M&M (L85) where only 29 additional isolates are mentioned. The reclassification of 2 strains of B. vini is not yet known in the introduction.

-L118: clarify numbers between brackets

-L145: what are” surprising results”?

-L234-235: does the 29 ribosomal SSU and LSU pertain to one strain?

-L431: reclassification to Heyndrickxia also applies to B. vini

-L488: this is not related to post-pasteurization milk spoilage but solely to UHT or sterilization.

Author Response

Response to the Editor and to the Reviewer (microorganisms-1062551)

Dear Reviewer,

we would like to thank you for your valuable comments and encouragement to revise the manuscript microorganisms-1062551 by Fiedler et al. We have modified and improved the manuscript according to your suggestions.

Below, you will find major changes we made to the manuscript. Additionally, a detailed response to your suggestions. We hope that these are to your satisfaction and that the manuscript will now be found suitable for publication.

Major improvements as recommended by the reviewers:

  • change of species/taxon names of Bacillus species according to the recent reclassifications (Gupta et al 2020). We used the website http://lpsn.dsmz.de as reference for valid species names. For example, Bacillus sporothermodurans (Pettersson et al. 1996) was changed to Heyndrickxia sporothermodurans (Gupta et al. 2020). While those names have only recently been introduced, they are valid descriptions of the respective taxons and species. Hence, we updated the terms throughout the manuscript and figures and included a table, which provides an overview about previous and current species names for better comprehension.
  • The title was changed to “Taxonomic evaluation of the Heyndrickxia (basionym Bacillus) sporothermodurans group (H. sporothermodurans, H. vini, H. oleronia) based on whole genome sequences”
  • Spelling and punctuation have been improved

Detailed response:

We would like to thank you for your competent review. A detailed response to the comments can be found below.

Reviewer:

-an important remark is about the recent reclassification of members of this group to the new genus Heyndrickxia. Likewise, some other Bacillus species are used in the study which are also reclassified such as B. firmus to Cytobacillus, B. shackletonii to Margalitia, etc. The authors argue not to use this new names and only use valid species names because at the moment of writing, these reclassifications would have not yet been completed. However, even at the moment of writing, these new names were already valid and should therefore preferably be used in the text and trees to reflect the correct taxonomic classification of Bacillus species. The authors can use the website http://lpsn.dsmz.de   to check the correct names of all species included.

Response:

We agree with you. As suggested by the reviewer, we changed the species/taxon names of Bacillus species according to the recent reclassifications (http://lpsn.dsmz.de) consistently throughout the manuscript, as well as in all figures, tables and supplementary files

We furthermore added the following sentences and table to the introduction:

Lines 42-48: “Recently a new classification of Bacillus species was published by Gupta et al. (2020), which proposes a reclassification and renaming of, among others, Bacillus sporothermodurans [5]. Based on this valid taxonomic reclassification, the new names listed in Table 1 were used in this study.”

Old name (basionym)

Publication

New name

Publication

Bacillus sporothermodurans

Pettersson et al. 1996

Heyndrickxia sporothermodurans

Gupta et al. 2020

Bacillus vini

Ma et al. 2017

Heyndrickxia vini

Gupta et al. 2020

Bacillus oleronius

Kuhnigk et al. 1996

Heyndrickxia oleronia

Gupta et al. 2020

Bacillus shackletonii

Logan et al. 2004

Margalitia shackletonii

Gupta et al. 2020

Bacillus camelliae

Niu et al. 2018

Margalitia camelliae

Gupta et al. 2020

Bacillus ginsengihumi

Ten et al. 2007

Weizmannia ginsengihumi

Gupta et al. 2020

Bacillus fordii

Scheldeman et al. 2004

Siminovitchia fordii

Gupta et al. 2020

Bacillus firmus

Bredemann and Werner 1933

Cytobacillus firmus

Patel and Gupta 2020

Bacillus luciferensis

Logan et al. 2002

Gottfriedia luciferensis

Gupta et al. 2020

Reviewer:

-another important remark is that the authors included the not-validated species “B. mediterraneensis” (to be written between quotation marks) in the 16S rDNA tree, but not “B. obstructivus” which also belongs to the B. sporothermodurans clade according to Gupta et al. (2020). Furthermore, a genome is available for the latter not-validated species in NCBI database. It would be very valuable if this species (although not validated) could be included in (at least some) of the analyses.

Response:

The 16S rDNA phylogenetic tree has been recalculated with the latest data and species names. We have removed “B. mediterraneensis” from the 16S analyses (only type strains). Our decision to not include B. obstructivus in the analyses is based on the lack of validation as a new species (type strain) on the one hand. On the other hand, our previous analyses (dDDH and 1000 core gene phylogeny, data not shown in this study) indicated that B. obstructivus is not a new species, but belong to the species Heyndrickxia oleronia. In this regard, we do not agree with the use of B. obstructivus (performed by Gupta et al.) and thus refrained from using the genome in the present study.

For your interest: dDDH Bacillus_obstructivus_GCF_001887185.1_ vs. Bacillus_oleronius_DSM9356 -DDH estimate (GLM-based): 90.50% [88.3 - 92.4%] Probability that DDH > 70% (i.e., same species): 95.97% (via logistic regression).

Reviewer:

-in the abstract, it should be described that 2 B. vini [sic] isolates have been reclassified as B. sporothermodurans [sic]. In Results, this reclassification should be described as soon as possible.

Response:

We thank the reviewer for this comment and agree. We added the information as follows:

Line 22: “After sequence analysis, the two H. vini strains could be reclassified as H. sporothermodurans.”

Lines 212-213: ”The two presumptive H. vini strains (identification of the strains was based on previous 16S rRNA gene Sanger sequencing) could be reclassified as H. sporothermodurans (see below).”

Reviewer:

-Discussion is rather long and contains repetitions with Results, so Discussion can be reduced.

Response:

We delete the following part from the Discussion:

Formerly lines 446-457: “The mol% G+C content shows slight changes depending on the sequencing method (e.g. 35.75 mol% GC by short-read assembly when compared to 36.36 mol% GC by hybrid assembly for B. sporothermodurans DSM 10599T) and is therefore not a good basis for comparing species, if different techniques were used. When comparing core genome from the Roary pipeline of B. sporothermodurans, B. vini and B. oleronius, the species B. sporothermodurans shares more genes with B. vini (964 genes), than with B. oleronius (158 genes), while the core genome of B. vini and B. oleronius, encompasses 190 genes. This clearly indicates a more distant relationship of B. oleronius to the other two species at the genomic level. The observed anomaly (158 core genes for B. sporothermodurans + B. oleronius and 164 for B. sporothermodurans + B. oleronius + B. vini) was identified as a technical artifact in the blastp calculation (Roary pipeline). While the artefact had no significant influence on other results (Supplementary Table 1), it highlights the necessity of thorough manual examination of complex computational analyses.”

We closely evaluated the text again and, to our opinion, found no further paragraphs that could be deleted or shortened without compromising comprehension for a broader audience. We hope the reviewer agrees with the implemented changes.

Reviewer:

-Conclusion is rather long and can be condensed without giving specific details.

Response:

We deleted the following sentence from the conclusion:

Formerly lines 553-555: “While physiological characteristics of the described ecotype are yet to be investigated, the present study provides a profound basis for future experiments in this research area.”

Reviewer:

-L41: sometimes pink coloration of milk is observed as spoilage

Response:

We thank the reviewer for the interesting comment. However, we are not aware of any study describing a pink coloration of milk upon H. sporothermodurans contamination. We have therefore not been able to adopt the proposal, but should the reviewer insist on this point, we kindly ask the reviewer to provide us with the relevant literature reference for this phenomenon.

Reviewer:

-L75-76: 31 additional strains of B. sporothermodurans is confusing here in relation to M&M (L85) where only 29 additional isolates are mentioned. The reclassification of 2 strains of B. vini is not yet known in the introduction.

Response:

We have corrected the number to 29 in the introduction (L85) to reduce confusion. Together with the reclassification mentioned early in the results, we hope to provide a more comprehensible indication of the numbers of H. sporothermodurans strains used.

Reviewer:

-L118: clarify numbers between brackets

Response:

Thank you for the comment. We altered the text accordingly:

Lines 128-132: “Long sequence reads were obtained by library preparation with the MinION 1D Native DNA barcoding genomic DNA protocol (with the Oxford Nanopore Ligation Sequencing Kit [EXP-NBD104] and the Oxford Nanopore Native Barcoding Expansion Kit [SQK-LSK 109]) and by sequencing using an Oxford Nanopore MinION MK1B sequencer with a MinION Flow Cell (R9.4.1) and a Flow Cell Priming Kit (EXP-FLP002).”

Reviewer:

-L145: what are” surprising results”?

Response:

We have changed “surprising results” to “illogical results”.

Reviewer:

-L234-235: does the 29 ribosomal SSU and LSU pertain to one strain?

Response:

Yes. But due to low relevance, we have removed this sentence completely.

Reviewer:

-L431: reclassification to Heyndrickxia also applies to B. vini

Response:

We have updated all strain names to the valid status.

Reviewer:

-L488: this is not related to post-pasteurization milk spoilage but solely to UHT or sterilization.

Response:

The technological delimitations are sometimes a bit ambiguous, but we fully agree with you and deleted “post-pasteurization”.

Lines 492-494: “A remarkable observation from a previous study done with a limited set of isolates showed that one clone, termed HRS, was involved in the majority of the cases of UHT milk spoilage.”

Submission Date

18 December 2020

Date of this review

08 Jan 2021 15:29:55

Reviewer 2 Report

This reviewer appreciate the work did by the authors and only minor change in editing the text is need it

Author Response

Response to the Editor and to the Reviewer (microorganisms-1062551)

Dear Reviewer,

we would like to thank you for your encouragement to revise the manuscript microorganisms-1062551 by Fiedler et al.

Below, you will find major changes we made to the manuscript. Additionally, a detailed response to your suggestions.

Major improvements as recommended by the reviewers:

  • change of species/taxon names of Bacillus species according to the recent reclassifications (Gupta et al 2020). We used the website http://lpsn.dsmz.de as reference for valid species names. For example, Bacillus sporothermodurans (Pettersson et al. 1996) was changed to Heyndrickxia sporothermodurans (Gupta et al. 2020). While those names have only recently been introduced, they are valid descriptions of the respective taxons and species. Hence, we updated the terms throughout the manuscript and figures and included a table, which provides an overview about previous and current species names for better comprehension.
  • The title was changed to “Taxonomic evaluation of the Heyndrickxia (basionym Bacillus) sporothermodurans group (H. sporothermodurans, H. vini, H. oleronia) based on whole genome sequences”
  • Spelling and punctuation have been improved

Detailed response:

We would like to thank you for your positive review.

Reviewer 3 Report

The manuscript written by Fiedler et al. reported a taxonomic evaluation of B. sporothermodurans bacteria based on the whole genome sequences. In this paper an interestingly approach was performed to increase the availability of genomic sequences, that consists in the whole genome comparison, coupled with the single nucleotide polymophisms and the phylogenomic analyses.
According to their data, a more detailed genomic resolution was achieved, utilizing an hybrid assembly strategy that has led to generate a single chromosome instead of several contigs. The article is well written, each analysis is well determined and worth to be published in Microorganisms. There are some minor comments.

- If the specie Bacillus is abbreviated as suggested in line 37, it should be also reported shorted in all the manuscript (line 39, 49, 55, 74)
-Table 4, I suggest to move “Bacillus sporothermodurans DSM 10599T” to the first column header.

Author Response

Response to the Editor and to the Reviewer (microorganisms-1062551)

Dear Reviewer,

we would like to thank you for your valuable comments and encouragement to revise the manuscript microorganisms-1062551 by Fiedler et al. We have modified and improved the manuscript according to your suggestions.

Below, you will find major changes we made to the manuscript. Additionally, a detailed response to your suggestions. We hope that these are to your satisfaction and that the manuscript will now be found suitable for publication.

Major improvements as recommended by the reviewers:

  • change of species/taxon names of Bacillus species according to the recent reclassifications (Gupta et al 2020). We used the website http://lpsn.dsmz.de as reference for valid species names. For example, Bacillus sporothermodurans (Pettersson et al. 1996) was changed to Heyndrickxia sporothermodurans (Gupta et al. 2020). While those names have only recently been introduced, they are valid descriptions of the respective taxons and species. Hence, we updated the terms throughout the manuscript and figures and included a table, which provides an overview about previous and current species names for better comprehension.
  • The title was changed to “Taxonomic evaluation of the Heyndrickxia (basionym Bacillus) sporothermodurans group (H. sporothermodurans, H. vini, H. oleronia) based on whole genome sequences”
  • Spelling and punctuation have been improved

Detailed response:

We would like to thank you for your review and have implemented your comments as follows.

Reviewer:

- If the specie Bacillus is abbreviated as suggested in line 37, it should be also reported shorted in all the manuscript (line 39, 49, 55, 74)

Response:

We agree with the suggested changes in principle (L37, L55 and L74). However, due to the general changes in species names, we have not exploited all the possibilities of abbreviations. For sentence beginnings and sporadically used genera, we preferred the complete name instead of abbreviations. We hope that this will make the content clearer.

Reviewer:

-Table 4, I suggest to move “Bacillus sporothermodurans DSM 10599T” to the first column header.

Response:

We thank the reviewer for this comment which is absolutely correct. We moved the type strains to the first and second line.

Submission Date

18 December 2020

Date of this review

29 Dec 2020 16:24:47

Round 2

Reviewer 1 Report

I thank the authors for their valuable revisions.

I have the following last remarks:

-basionym should be basonym

-a suggestion is to include the observation of "B. obstructivus" as belonging to H. oleronia in the manuscript because the authors have the necessary data.

-concerning the pink coloration of UHT-milk which sometimes appears after contamination with HRS spores, I acknowledge it is not easy to find a reference for this observation. It has been originally described by Lembke, F. (1995) Highly Heat-Resistant spores in UHT-milk. Bulletin of the International Dairy Federation 302, 60–61. It is not a standard publication, so I leave to the authors to decide to use it.

-species names in Table 1 should be in italics

Author Response

Response to the Reviewer (microorganisms-1062551)

Reviewer 1

Open Review

We would like to thank you for your competent and comprehensive review. A detailed response to the comments can be found below.

Reviewer:

-basionym should be basonym

Response:

We agree with you. Very thoughtful, thank you. As suggested by the reviewer, we changed basionym to basonym as following:

Manuscript:

L3 Title: “Taxonomic evaluation of the Heyndrickxia (basonym Bacillus) sporothermodurans group (H. sporothermodurans, H. vini, H. oleronia) based on whole genome sequences”

L47-48 Table 1

L472 Table 4

Reviewer:

-a suggestion is to include the observation of "B. obstructivus" as belonging to H. oleronia in the manuscript because the authors have the necessary data.

Response:

We thank you for considering this data worthy of publication. However, we have decided not to include this data. The publication by Gupta et al. marks B. obstructivus as not validly published, so no correction from us is “needed”. In addition, this issue would be quite separate from the topics covered in the paper, and we would like to avoid additional confusion for the reader, as this paper is quite complex.

But, please feel free to contact us for detailed data and potential use. (dDDH=90.50%, ANIb=98.3%, etc)

Reviewer:

-concerning the pink coloration of UHT-milk which sometimes appears after contamination with HRS spores, I acknowledge it is not easy to find a reference for this observation. It has been originally described by Lembke, F. (1995) Highly Heat-Resistant spores in UHT-milk. Bulletin of the International Dairy Federation 302, 60–61. It is not a standard publication, so I leave to the authors to decide to use it

Response:

We thank the reviewer for this relevant information and the option of choice. However, we have not yet been able to get this publication online. However, through contact with Dr. Hammer (involved in the initial description and published together with Dr. Lembke) we were able to get the original article in print. The following part is relevant here:

Bulletin of the IDF 302, HIGHLY HEAT-RESISTANT SPORES IN UHT-MILK, F. Lembke

"Sensory and/or physical changes (for example coagulation, proteolysis) normally do not occur at least during the ordinary storage time at ambient temperatures, probably as a result of limited growth of the strictly aerobic strictly aerobic HRS in the product never exceeding log105. However, coagulated and pink-coloured milk has been reported in containers with a low oxygen barrier (for example plastic bottles)."

This description is not sufficient for us to describe a general phenomenon. According to personal discussions with Dr. Hammer, this colouration has never been observed in contaminated samples. This phenomenon was probably reported from Italy and adopted by Dr. Lembke. My scientific conscience does not permit the duplication of this statement.

Reviewer:

-species names in Table 1 should be in italics

Response:

Absolute, and we formatted all species names in Table 1.